# A Small Object Detection Algorithm for Traffic Signs Based on Improved YOLOv7

**DOI:** 10.3390/s23167145

**Published:** 2023-08-13

**Authors:** Songjiang Li, Shilong Wang, Peng Wang

**Affiliations:** 1College of Computer Science and Technology, Changchun University of Science and Technology, Changchun 130022, China; lsj@cust.edu.cn (S.L.); wsl@mails.cust.edu.cn (S.W.); 2Chongqing Research Institute, Changchun University of Science and Technology, Chongqing 401120, China

**Keywords:** deep learning, traffic sign detection, YOLOv7, small object detection, ACmix, computer vision

## Abstract

Traffic sign detection is a crucial task in computer vision, finding wide-ranging applications in intelligent transportation systems, autonomous driving, and traffic safety. However, due to the complexity and variability of traffic environments and the small size of traffic signs, detecting small traffic signs in real-world scenes remains a challenging problem. In order to improve the recognition of road traffic signs, this paper proposes a small object detection algorithm for traffic signs based on the improved YOLOv7. First, the small target detection layer in the neck region was added to augment the detection capability for small traffic sign targets. Simultaneously, the integration of self-attention and convolutional mix modules (ACmix) was applied to the newly added small target detection layer, enabling the capture of additional feature information through the convolutional and self-attention channels within ACmix. Furthermore, the feature extraction capability of the convolution modules was enhanced by replacing the regular convolution modules in the neck layer with omni-dimensional dynamic convolution (ODConv). To further enhance the accuracy of small target detection, the normalized Gaussian Wasserstein distance (NWD) metric was introduced to mitigate the sensitivity to minor positional deviations of small objects. The experimental results on the challenging public dataset TT100K demonstrate that the SANO-YOLOv7 algorithm achieved an 88.7% mAP@0.5, outperforming the baseline model YOLOv7 by 5.3%.

## 1. Introduction

With the continuous advancements in computer, communication, and intelligent signal processing technologies, sophisticated advanced driver assistance systems (ADAS) and autonomous driving systems (ADS) gain increasing popularity. Traffic sign detection technology, as an integral component of these systems, plays a vital role in assisting with driving and enhancing road safety. This technology recognizes traffic signs, including speed limit signs, prohibition signs, and warning signs, in images or videos, providing essential information to drivers and autonomous vehicles. It ensures the smooth operation of the road network, effectively reducing or preventing traffic accidents, and to some extent, ensuring road travel safety. However, the current traffic sign detection technology continues to encounter challenges in detecting small target objects. Small target traffic signs usually occupy a small portion of real-world road scenes, posing difficulties for detectors to extract relevant features. As shown in Figure 1, within a traffic scene image captured by a real vehicular camera with a resolution of 2048 × 2048 pixels, we can identify two small traffic sign targets, both measuring only 36 × 36 pixels, accounting for approximately 0.03% of the entire image size. Due to their low resolution and limited information content, these small targets are susceptible to being overlooked or falsely detected during the detection process, presenting a significant challenge for small object detection tasks. Moreover, the simultaneous presence of small target objects alongside objects of different sizes within the same image, displaying significant scale variations, results in missed or false detections, consequently decreasing the accuracy of the detection process. Complex road scenarios, which encompass lighting variations, object occlusions, and adverse weather conditions, further exacerbate the challenge of detecting small target traffic signs. In recent years, many scholars have proposed theoretical frameworks and methodologies to enhance the performance of small object detection in traffic sign recognition. Method [1] introduced an end-to-end deep learning model for detecting and recognizing traffic signs in high-resolution images, significantly improving accuracy. However, this method is not suitable for real-time detection. Method [2] proposed an adaptive image enhancement algorithm to improve image quality under complex lighting conditions, and introduced a novel and lightweight attention block named the feature difference (FD) model for traffic sign detection and recognition, achieving promising results. However, the scale of the research dataset was relatively small, leading to potential randomness in the experimental results. Method [3] employed a technique to fuse low-level features into high-level features, thereby enhancing the detection performance of small objects in single shot multibox detector (SSD). It achieved favorable results by enhancing effective channel features and suppressing ineffective channel features for target detection. However, for real-time applications, there is still significant room for improving accuracy. 

In response to these challenges, we introduce the SANO- YOLOv7 algorithm for traffic sign detection, which builds upon the YOLOv7 framework. Our algorithm builds upon the latest version of Yolov7 as the baseline model, integrating improvements and modifications into the YOLOv7 network architecture. To assess the performance of our model, we opted for the TT100K [4] dataset, which comprises a substantial collection of small traffic sign samples captured by vehicular cameras in real-world traffic scenarios. These small traffic signs, present in diverse natural settings, provide a comprehensive representation of real-world traffic conditions. This dataset is highly suitable for our research purposes. We assess the performance of the enhanced algorithm on the TT100K dataset, demonstrating favorable results in terms of detection and classification accuracy, as well as improved overall model capabilities.

The main contributions of this paper are summarized as follows:Based on the original network, a fourth feature prediction scale was added into the YOLOv7 network architecture to effectively utilize shallow features for the precise detection of small target objects.Following the first feature layer output, we introduced a self-attention and convolutional mixture module (ACmix) that enables the model to learn from large-scale feature maps of low-level outputs in the backbone network. The ACmix mechanism utilizes convolutional and self-attention channels to capture additional features, thereby enhancing the detection capability of small target objects.To enhance the feature extraction capability of the convolutional module and capture more contextual clues, we introduced omni-dimensional dynamic convolution (ODConv) as a novel feature extraction module called ODCBS. ODCBS enables the parallel learning of convolutional kernel features across all four dimensions of the convolutional kernel space, thus capturing more comprehensive contextual information.In order to improve the detection performance of small objects, the normalized Gaussian Wasserstein distance (NWD) metric was introduced in both the non-maximum suppression (NMS) and the loss function. The NWD metric addresses the sensitivity of IoU to slight positional deviations of small objects, resulting in a significant enhancement in the accuracy of small object detection.

The rest of this paper is organized as follows: Section 2 offers a concise overview of the relevant literature and methodologies about traffic sign detection and small object detection. Section 3 presents our research approach. Section 4 describes the experimental results. Lastly, in Section 5, we summarize the work and draw conclusions.

## 2. Related Work

### 2.1. Traffic Sign Detection

#### 2.1.1. Traditional Traffic Sign Detection

Traditional traffic sign detection algorithms can be classified into three main categories: color-based, shape-based, and machine learning-based.

Color-based traffic sign detection typically involves the process of color-based image segmentation, which is followed by sending the segmented regions of interest to a classifier to determine whether they represent traffic signs. In the 1980s, Akatsuka H et al. [5] introduced a traffic sign detection method that utilized the RGB color space model. The authors employed thresholding segmentation algorithms with various threshold ranges for colors such as red, yellow, and blue, resulting in successful detection of traffic signs by leveraging color recognition. Yang et al. [6] converted the input color image into a probability map and detected traffic signs by the maximum stable extreme value region method. KiranC.G et al. [7] proposed an algorithm for traffic sign image segmentation that was based on color enhancement lookup tables (LUTs). The algorithm initially mapped the image from the RGB color space to the HSI color space and subsequently enhanced the red, yellow, and blue colors by adjusting their H and S values. This method effectively enabled the localization of various types of traffic signs in images. While color-based methods enhanced real-time operation and accuracy, detection outcomes could be susceptible to interference due to challenging conditions such as lighting and rain. Furthermore, in the case of faded or damaged traffic signs, the algorithm’s detection performance could be significantly compromised, resulting in missed detections and false positives.

Shape-based traffic sign detection begins with edge detection and shape detection to extract shape features, followed by detection using a classifier. Reference [8] improved the robustness of the detection method by limiting the Hough transform in specific regions for detecting circular signs and using the line Hough transform for detecting triangular signs. In 2013, Boumediene et al. [9] introduced an effective approach to detect triangular traffic signs utilizing the rotational symmetry line detection (RSLD) algorithm. This algorithm converted the detection of triangles into the detection of simple line segments, enabling the identification of warning and yield traffic signs regardless of their distance. Moreover, reference [10] comprehensively addressed the shape features of triangles, circles, and squares. It employed connected components for shape recognition and eliminated non-traffic sign shapes from the image, resulting in a high level of detection performance. Although shape-based detection methods are effective in mitigating interference caused by color fading, they may experience a considerable decrease in detection performance when traffic signs undergo deformation or changes in the detection viewpoint occur.

Machine learning-based methods extract features from regions of interest in traffic sign images and utilize more effective classifiers for traffic sign detection. In 2005, Dalal [11] proposed the histogram of oriented gradients (HOG) algorithm, which utilized gradient orientation histograms to describe location-specific feature information in images, normalized them, and effectively detected the local data of target features in images. Subsequently, Ardianto et al. [12] implemented a traffic sign detection system using HOG and support vector machines (SVMs), achieving an accuracy of 91%. Chen et al. [13] introduced a traffic sign classifier based on AdaBoost technology, combined with support vector regression to enhance the detection performance.

In general, these traditional traffic sign detection algorithms possess strengths and limitations in different aspects. However, in recent years, with the advancement of deep learning, deep learning-based methods have demonstrated superior performance in the field of traffic sign detection.

#### 2.1.2. Traffic Sign Detection Based on Deep Learning

Traditional methods for traffic sign detection and recognition have yielded positive outcomes. However, these methods require manual feature extraction by researchers and are susceptible to various factors such as lighting, angles, deformations, and complex backgrounds. Consequently, they exhibit limited generalization ability and struggle to meet the demands of traffic sign recognition tasks in natural environments. In contrast, deep learning-based detection methods excel in extracting profound features of traffic signs and exhibit superior abilities to detect a broader range of categories, demonstrating higher levels of generalization and detection capabilities. At present, convolutional neural network (CNN)-based object detection algorithms encompass two primary types: region-based detection methods and regression-based detection methods.

Region-based detection methods, also known as two-stage methods, divide the detection task into two stages. The first stage involves extracting candidate regions in the image, usually referred to as “region proposal”, using methods such as selective search or region proposal network (RPN). The second stage involves classifying and localizing these candidate regions. Typically, a CNN is used for feature extraction, followed by a classifier and a regressor to classify and refine the positions of the candidate regions. The entire process divides the image into multiple smaller regions for processing and ultimately obtains the detection results of all targets. Representative object detection methods include R-CNN [14], Fast R-CNN [15], Faster R-CNN [16], and Mask R-CNN [17] models. In the field of traffic sign detection, Zhang et al. [18] proposed a cascaded R-CNN with multi-scale attention and imbalance sample handling, significantly improving the detection performance of small-sized traffic signs. Cao et al. [19] used a parallel fusion feature extraction network HRNet to improve the feature extractor of the Faster R-CNN model, achieving good detection accuracy and robustness on the TT100K traffic sign dataset. Shao et al. [20] proposed a region proposal algorithm based on simplified Gabor wavelet (SGW) and maximally stable extremal regions (MSERs) as the first step, obtaining prior information of region proposals and using it to improve Faster R-CNN for the better detection of small-sized traffic sign targets. Although these methods perform well in accuracy, they are relatively slow in detection speed and lack real-time capabilities.

Regression-based detection methods, also known as one-stage methods, transform the object detection task into a regression problem, directly detecting objects from the image without the need for prior generation of candidate boxes. Consequently, they offer the advantage of faster detection speed. Representative algorithms include You Only Look Once (YOLO [21]), single shot multibox detector (SSD [22]), and RetinaNet [23]. In the field of traffic sign detection, a study [24] addressed the issue of low accuracy in small object recognition by improving the YOLO network. By integrating spatial pyramid pooling into the YOLOv3 network, it comprehensively learned multi-scale features and effectively enhanced the detection accuracy of traffic signs to 91%. Jiang et al. [25] introduced depth-wise separable convolution to the feature extraction layer of YOLOv3, ensuring high accuracy while significantly reducing the model’s parameters and computational complexity. Jin et al. [3] proposed an improved SSD algorithm that integrates and enhances low-level features into high-level features, reinforcing impactful features in different channels and suppressing ineffective features, achieving promising results on traffic sign datasets. Wang et al. [26] presented an enhanced feature pyramid network model called AF-FPN, which incorporates the adaptive attention module (AAM) and feature enhancement module (FEM) to reduce information loss in the feature map generation process and enhance the representation capability of the feature pyramid. By replacing the original feature pyramid network in YOLOv5 with AF-FPN, the model’s recognition capability for traffic signs was improved. Yao et al. [27] introduced an improved YOLOv4-Tiny for real-time traffic sign detection. This method proposed an adaptive feature pyramid network (AFPN) to adaptively fuse feature layers at two different scales. Additionally, two receptive field blocks (RFBs) were added after the two feature layers of the backbone network to enhance the feature extraction capability of the backbone network. The experimental results demonstrated the effective improvement of traffic sign detection performance achieved by the enhanced method.

### 2.2. Small Object Detection

The definition of small objects can be classified into two categories: relative scale and absolute scale. The relative scale is determined by the proportion of target pixels within the image. According to the SPIE organization, objects with a pixel ratio to the total image pixels less than 0.12% are considered small objects. Furthermore, small objects can also be defined based on the aspect ratio of the object’s bounding box in relation to the image’s width and height. Typically, objects with an aspect ratio smaller than 0.1 are categorized as small objects. On the other hand, absolute scale defines small objects based on their absolute pixel count, and this definition can vary depending on the dataset. The COCO dataset defines small objects as those with a resolution smaller than 32 × 32 pixels. In contrast, the TinyPerson dataset classifies objects ranging from 20 to 32 pixels in size as small objects. 

Detecting small objects poses a significant challenge in computer vision. Firstly, small objects usually occupy a smaller region in an image, with sizes considerably smaller than medium to large objects. Small objects have limited feature information due to their small size, making it challenging to extract enough features from the limited number of pixels for accurate detection. Additionally, small objects frequently appear in complex backgrounds, displaying low contrast with the surrounding environment and lacking distinct edges and color contrasts. Lin et al. [28] proposed the feature pyramid network (FPN), which constructs a feature pyramid to offer multi-scale feature representations for small objects. This approach tackles the issue of limited small object information in deep features, thereby improving the detection capability of small objects. However, FPN exhibits suboptimal performance in handling unreliable region details at lower levels and information loss during the downsampling process. Additionally, it faces challenges in dealing with highly imbalanced target size distributions, particularly when extremely small and large objects coexist. Reference [29] employed generative adversarial learning to map the features of small, low-resolution objects to the same features as high-resolution objects, achieving comparable detection performance with larger-sized objects. However, generative adversarial networks may face challenges, such as training instability and mode collapse, as the competition between the generator and discriminator can hinder model convergence or the generation of diverse samples. Reference [30] proposed an effective selective context network (ESCNet) to address the shortcomings of the single shot multibox detector (SSD) network in context exploration. This network introduced an enhanced context module for extracting contextual information from original-scale, small-scale, and large-scale contextual information. Additionally, ESCNet incorporated a triple attention module, consisting of global-level, channel-level, and spatial-level attention, to fuse contextual information and selectively refine features. Nevertheless, ESCNet continues to face difficulties in handling background confusions, resulting in false positives during the detection of small objects in complex backgrounds. Deng et al. [31] proposed an extended pyramid network to remedy the problem of small object detection, where a layer specialized for small objects is generated by the FPN-like framework. A novel feature texture transfer module is embedded in the FPN-like framework to efficiently capture more regional details for the extended pyramid level, greatly enhancing the detection capability for small targets.

### 2.3. YOLOv7 Network Structure

YOLOv7 [32] is the latest and most advanced object detection model in the YOLO series. The pioneering YOLOv1 [21] was introduced in 2015, streamlining object detection by performing object localization and classification in a single forward pass, effectively addressing the slow inference speed of two-stage detection networks. Building upon its predecessor, YOLOv3 [33] further improved performance through multi-scale feature fusion and the introduction of the Darknet-53 residual module. Subsequently, YOLOv4 [34] and YOLOv5 have added many tricks based on version 3. In 2022, YOLOv7 emerged, advancing upon the YOLOv5 network by incorporating several strategies, including the extended efficient long-range attention network (E-ELAN), model scaling based on concatenation-based models, and convolution reparameterization. These enhancements notably improved the network’s reasoning speed and accuracy, positioning YOLOv7 as the most advanced object detector, achieving remarkable results in the range from 5 FPS to 160 FPS. The experimental findings have demonstrated that YOLOv7 outperforms YOLOR [35], YOLOX [36], YOLOv5, DETR [37], Deformable DETR [38], and numerous other object detectors. Therefore, we chose YOLOv7 as our baseline model over other methods. The YOLOv7 framework consists of three core models: YOLOv7-tiny, YOLOv7, and YOLOv7-W6, each designed to cater to different execution environments (edge GPU, regular GPU, and cloud GPU).

As illustrated in Figure 2, the YOLOv7 architecture comprises four main components: the input, backbone, neck, and head.

The input layer employs three techniques: Mosaic data augmentation, adaptive anchor box calculation, and adaptive image scaling. The preprocessed color image is uniformly resized to 640 × 640 pixels, satisfying the input size requirement of the backbone network. 

The backbone module consists of several CBS convolution layers, ELAN convolution layers, and MP1 convolution layers. CBS includes a convolution layer, a batch normalization (BN) layer, and a SiLU activation function, facilitating the extraction of image features at various scales. The ELAN module comprises multiple convolutional modules and regulates the shortest and longest gradient paths to facilitate improved learning and convergence. The MP1 module contains a max pooling layer and several convolutional modules. It employs upper and lower branches to downsample the feature maps, reducing the image’s dimensions and channel numbers by half. Subsequently, feature fusion is performed to enhance the model’s capability for feature extraction.

The neck network employs a path aggregation feature pyramid network (PAFPN) structure. It consists of several CBS blocks, along with the incorporation of a spatial pyramid pooling and convolutional spatial pyramid pooling (SPPCSPC) structure, ELAN-H, and MP2. The SPPCSPC structure enhances the network’s perceptual field by integrating a convolutional spatial pyramid (CSP) structure within the spatial pyramid pooling (SPP) structure, along with a large residual edge to optimize feature extraction. The ELAN-H layer, which combines multiple feature layers based on ELAN, further improves feature extraction. The MP2 block has a similar structure to the MP1 block but with a slight modification in the number of output channels. The neck network achieves the integration of high-resolution and high-semantic information by merging high-level features with low-level features.

The head network employs the REP module to adjust the channel sizes of output features of different scales. It converts these features into bounding box, class, and confidence information. A convolutional layer serves as the detection head for downsampling, enabling the detection of objects at multiple scales, including large, medium, and small targets.

## 3. Materials and Methods

### 3.1. Small Object Detection Structure 

The original YOLOv7 network utilizes three feature maps of varying scales to detect objects of different sizes. For an input image size of 640 × 640 pixels, the feature map sizes are 80 × 80 × 255, 40 × 40 × 255, and 20 × 20 × 255, corresponding to detection box sizes of 8 × 8 pixels, 16 × 16 pixels, and 32 × 32 pixels, respectively. However, in the TT100K dataset, numerous targets have sizes smaller than 8 × 8 pixels, resulting in inadequate detection performance and the possibility of false positives or missed detections within the original network architecture. To address this issue, a dedicated small-target detection layer is introduced to enhance the model’s performance. Following the generation of an 80 × 80 scale feature map from the FPN module, an upsampling operation is conducted to acquire a 160 × 160 feature map. Subsequently, the fused feature map, obtained by integrating the 160 × 160 feature map from the shallow layers of the backbone module, is directly fed into the prediction module, creating a specialized prediction layer specifically designed for small targets. This enhancement effectively improves the accuracy of small target detection while introducing only a minimal increase in the parameter count. The specific improvement is illustrated in the blue box in Figure 3.

### 3.2. ACmix

To effectively detect small traffic sign targets, two primary challenges must be overcome [39,40]. Firstly, small objects have fewer pixels and weaker feature representation, resulting in limited representative feature information. Secondly, in complex background environments, small objects are prone to being confused with the background, making it challenging to extract their distinctive feature information through standard feature extraction methods. To address these challenges, an attention mechanism can be integrated into the feature extraction process. By doing so, the attention mechanism assists the network in focusing on specific regions, learning distribution patterns, recalibrating attention, and emphasizing significant locations. This integration enhances the network’s capability for detecting small targets effectively. Traditional convolution utilizes shared weights across the feature map to aggregate information from local receptive fields. Weight-sharing reduces the number of parameters and improves training efficiency. Additionally, it allows for capturing local patterns and structural information in the input features. On the other hand, self-attention modules compute attention weights dynamically by applying a weighted average operation based on the context of input features and using similarity functions between related pixel pairs. This flexibility enables the attention module to adaptively focus on various regions, making it adept at capturing internal correlations within the data or features. Pan et al. [41] proposed a hybrid attention mechanism called ACmix module in CVPR 2022, which effectively combines the advantages of traditional convolution and self-attention modules. The structure of ACmix is illustrated in Figure 4.

ACmix is divided into two stages. In the initial stage, the H × W × C features are projected through three 1 × 1 × C convolutions to restructure the input features into N segments, resulting in a subset of sub-features comprising 3 × N feature maps. In the subsequent stage, the feature subset obtained from the previous stage is fed into two separate branches. The upper branch follows a convolutional path with a kernel size of k, collecting information from local receptive fields. A lightweight fully connected layer is then used to convert the features into k^2^ feature maps. These generated features undergo shift, aggregation, and convolution operations, resulting in H × W × C feature maps. Meanwhile, the lower branch incorporates global information using self-attention. The intermediate features are divided into N groups, each consisting of 3 feature maps: query, key, and value. These grouped features are processed using a multi-headed self-attention model. Afterwards, the features undergo additional processing through shift, aggregation, and convolution to yield H × W × C feature maps. Finally, the outputs from both branches are weighted and summed, with the weights determined by two learnable scalars, as indicated in Equation (1):(1)$$F_{out}=\alpha F_{conv}+\beta F_{att}$$
where $F_{out}$ represents the final output of the pathway; $F_{att}$ represents the output of the self-attention branch; $F_{conv}$ represents the output of the convolution attention branch. In this paper, both *α* and *β* are set to 1.

The outputs of the two branches are merged, simultaneously considering global and local features, thereby enhancing the network’s detection performance for small targets.

### 3.3. ODCBS Module

Traditional convolution, also referred to as static convolution, is a fundamental operation widely employed in convolutional neural networks (CNNs). It applies a fixed kernel size and stride to convolve the input feature maps. However, the weights of traditional convolution are fixed and remain unchanged regardless of variations in the input data. To enhance the performance of the network, it becomes necessary to employ deeper and wider network structures while incorporating strategies like residual connections and multi-scale fusion. Nonetheless, traditional convolution employs the same weight parameters for convolutional operations at every position, resulting in parameter redundancy. Moreover, the fixed kernel size limits the receptive field, thereby constraining the model’s capability to capture features at different scales and levels. Furthermore, traditional convolution solely considers local pixel relationships and fails to capture global contextual information. To address these concerns, dynamic convolution has been proposed, which introduces learnable kernel weights. Unlike traditional convolution, the kernel weights of dynamic convolution adaptively learn based on the content of the input data. This adaptive nature enables dynamic convolution to better align with the feature representation requirements of diverse input data, reducing parameter redundancy, enhancing parameter efficiency, and possessing improved multi-scale perception and representation capabilities of global contextual information. The operation of dynamic convolution can be defined as Formula (2):(2)$$y=\left(\alpha_{w1}W_{1}+\ldots +\alpha_{wn}W_{n}\right)\times x$$
where *x* and *y* denote input features and output features, respectively; $W_{i}$ denotes the $i^{th}$ convolutional kernel consisting of $c_{out}$ filters $W_{i}^{m}\in R^{k\times k\times c_{in}}$, m = 1, …, $c_{out}$; $\alpha_{wi}\in R$ is the attention scalar for weighting $W_{i}$, which is computed by an attention function $\pi_{wi}(x)$ conditioned on the input features.

This paper introduces omni-dimensional dynamic convolution (ODConv [42]), which employs a novel multi-dimensional attention mechanism and parallel strategies to learn complementary attention in all four dimensions of the kernel space for any convolutional layer. The four distinct attention mechanisms encompass the input channel count of the convolutional kernel, the receptive field of the kernel itself, the output channel count of the kernel, and the number of convolutional kernels. These attention mechanisms mutually complement one another, and their application to the convolutional kernel enhances the feature extraction capability of CNNs. The output of ODConv can be represented using Formula (3).
(3)$$y=\left(\alpha_{w1}⊙\alpha_{f1}⊙\alpha_{c1}⊙\alpha_{s1}⊙W_{1}+\ldots +\alpha_{wn}⊙\alpha_{fn}⊙\alpha_{cn}⊙\alpha_{sn}⊙W_{n}\right)\times x$$
where *x* and *y* represent the input features and output features, respectively. $\alpha_{wi}$ is the attention scalar for the $i^{th}$ convolution kernel, while $\alpha_{si}$, $\alpha_{ci}$ and $\alpha_{fi}$ represent the attention scalars along the spatial, input channel, and output channel dimensions, respectively. ⊙ represents the element-wise multiplication along different dimensions of the kernel space. The calculation process of ODConv is depicted in Figure 5.

The ODCBS module comprises three components. The first component is ODConv, which facilitates comprehensive feature extraction throughout the kernel space. The second component is batch normalization, which mitigates the problems of gradient vanishing and exploding. The final component is the SiLU activation function, which promotes smooth gradient flow and further improves the stability of the model. The structure of the ODCBS module is depicted in Figure 6.

### 3.4. Normalized Gaussian Wasserstein Distance

In the field of object detection, metrics based on intersection over union (IoU), such as IoU itself and its extensions, are highly sensitive to small positional deviations in tiny objects. As depicted in Figure 7, each grid represents a pixel, where box A denotes the ground-truth bounding box, and boxes B and C represent predicted bounding boxes with diagonal deviations of 1 pixel and 4 pixels, respectively. The observation reveals that for objects of regular size, minor positional variations typically have a negligible impact on the IoU. However, in the case of extremely small targets, even a slight positional deviation can result in a significant decrease in the IoU. This scenario leads to numerous predicted boxes having IoU values below the predefined threshold, potentially causing false detections and significantly impacting detection performance.

To address this concern, Wang et al. [43] proposed a novel metric, named normalized Gaussian Wasserstein distance (NWD), for the detection of small objects. In this approach, bounding boxes are initially represented as two-dimensional Gaussian distributions, and the dissimilarity between the predicted and ground-truth objects is quantified by assessing the similarity of their respective Gaussian distributions. The computation formula for the Wasserstein distance between two two-dimensional Gaussian distributions, P and Q, is as follows:(4)$$W_{2}^{2}\left(\mu_{{}_{1}},\mu_{2}\right)={\left\|m_{1}-m_{2}\right\|}_{2}^{2}+Tr\left(\Sigma_{1}+\Sigma_{2}-2{\left(\Sigma_{2}^{1/2}\Sigma_{1}\Sigma_{2}^{1/2}\right)}^{1/2}\right)$$ furthermore, for Gaussian distributions $N_{a}$ and $N_{b}$ which are modeled from bounding boxes A = $\left(cx_{a},cy_{a},w_{a},h_{a}\right)$ and B = $\left(cx_{b},cy_{b},w_{b},h_{b}\right)$, Equation (4) can be further simplified as:(5)$$W_{2}^{2}(N_{a},N_{b})={\left\|\left({\left[cx_{a},cy_{a},\frac{w_{a}}{2},\frac{h_{a}}{2}\right]}^{T},{\left[cx_{b},cy_{b},\frac{w_{b}}{2},\frac{h_{b}}{2}\right]}^{T}\right)\right\|}_{2}^{2}$$ however, $W_{2}^{2}(N_{a},N_{b})$ is a distance metric that cannot be directly used as a similarity measure. Therefore, it is necessary to normalize this distance by transforming $W_{2}^{2}(N_{a},N_{b})$ into a value between 0 and 1, thereby obtaining the normalized Gaussian Wasserstein distance (NWD):(6)$$NWD(N_{a},N_{b})=\exp\left(-\frac{\sqrt{W_{2}^{2}(N_{a},N_{b})}}{C}\right)$$

In this paper, we introduced the normalized *NWD* metric in both the non-maximum suppression (NMS) and the loss function, with the aim of mitigating the sensitivity of IoU to small object displacement.

#### 3.4.1. NWD-NMS

During the prediction stage of object detection, multiple candidate bounding boxes may be generated for a single image, with each candidate box corresponding to a detected object. However, due to variations in the position, scale, and orientation of objects in the image, as well as the possibility of different candidate boxes corresponding to the same object, these candidate boxes may overlap with each other. If not properly handled, these overlapping candidate boxes can lead to multiple detections by the algorithm, resulting in increased false positives and decreased detection speed. To address this issue, the non-maximum suppression algorithm (NMS) can be used to filter out overlapping candidate boxes. The basic idea of the NMS algorithm is to retain the candidate box with the highest confidence for each class and remove other candidate boxes that have an intersection over union (IoU) with the selected box greater than a certain threshold. This method effectively eliminates redundant candidate boxes, reducing false positives and improving detection speed.

However, for small object detection, the IoU-based NMS algorithm can lead to false positive predictions and scale sensitivity, resulting in suboptimal performance. Therefore, in NMS, we adopt the normalized Gaussian Wasserstein distance (NWD) as a replacement for the original IoU as the measurement criterion. The specific procedure is as follows: first, the predicted bounding boxes are sorted in descending order of confidence. The bounding box with the highest confidence is selected as the reference box, and the NWD between this box and other boxes is calculated. Boxes with an NWD below a certain threshold are then removed. Next, the box with the second highest confidence is chosen as the new reference box, and the process is repeated until all predicted bounding boxes have been used as reference boxes. This approach ensures that excessively similar bounding boxes are not present, significantly reducing the computational load of predicting bounding boxes.

#### 3.4.2. NWD-Loss

The introduction of IoU-Loss aims to mitigate the performance gap between training and testing. However, in small object detection, when there is no overlap between the predicted bounding box A and the ground-truth bounding box B, or when bounding box A completely contains bounding box B (and vice versa), IoU-Loss fails to provide the necessary gradients for optimizing the network. To address this issue, The NWD-Loss is added in order to compensate for the limitations of the CIoU-Loss in small object detection. By retaining the CIoU-Loss, the algorithm achieves faster convergence in predicting bounding box localization and enhances the overall model performance. The computation formula for NWD-Loss is as follows:(7)$$L_{NWD}=1-NWD\left(N_{a},N_{b}\right)$$

### 3.5. The Proposed SANO-YOLOv7 Model

The structure of the SANO-YOLOv7 obtained by improving YOLOv7 is shown in Figure 8. Firstly, to improve the detection of small objects, a small object detection layer is added in the neck region, accompanied by an additional small object detection head. Secondly, an ACmix module, combining self-attention and convolutional mixing, is connected after the first feature layer. This module learns from the large-scale feature maps of the low-level outputs, capturing more features and enhancing sensitivity to small-scale targets while reducing the impact of noise. The ACmix mechanism is selectively applied to shallow layers to enhance small object detection without significantly increasing the computational complexity, thus avoiding resource redundancy when the improvement in detection accuracy is not substantial. Thirdly, the paper introduces omni-dimensional dynamic convolution and proposes the ODCBS module, replacing the original CBS module in the neck layer. The ODCBS module utilizes parallel strategies to learn attentions in all four dimensions of the kernel space, aiming to capture contextual information and prioritize traffic sign features. Finally, we introduce the normalized Gaussian Wasserstein distance (NWD) metric. Specifically, NWD-NMS replaces the original IoU-NMS post-processing method, and NWD-Loss is added into the loss function to mitigate the sensitivity of IoU loss to the positional deviations of small objects.

## 4. Experiments and Analysis of Results

### 4.1. Datasets

The algorithm employed in this study was trained and evaluated using the TT100K traffic sign dataset. The TT100K dataset, a collaborative effort between Tsinghua University and Tencent, encompasses a substantial number of road traffic signs in various complex environments and weather conditions, very small in size, which makes them extremely difficult to detect. The dataset comprises a total of 26,349 images, representing 221 distinct categories of traffic signs, with annotations available for 128 categories. The original street view panorama images have a resolution of 8192 × 2048 pixels, which were subsequently cropped to a final dataset size of 2048 × 2048 pixels. The training set consists of 6107 images, while the test set comprises 3073 images. Additionally, there are 7643 supplementary images. Figure 9 illustrates a collection of sample images extracted from the TT100K dataset.

Due to significant data imbalance among different categories in the TT100K dataset, it may impact the experimental results. To address this issue, we analyzed the number of various types of traffic signs in the dataset, and selected 42 categories of traffic signs with a number more than 100 to balance the sample discrepancy caused by different categories of traffic signs in the dataset. The selected traffic signs in the experiment can be categorized into three main types: warning signs, prohibition signs, and directional signs. The specific traffic signs chosen for the experiment are listed in Table 1. 

The distribution of the center points and sizes of the bounding boxes in the processed dataset is shown in Figure 10. Figure 10a illustrates the positional distribution of the center coordinates of the normalized bounding boxes after image resolution adjustment. In Figure 10b, the width and height represent the ratios of the bounding box width and height to the image width and height, respectively. From both figures, it is evident that the majority of the targets in the dataset are primarily concentrated in the central region. Moreover, a substantial number of small-sized objects are present. These characteristics render the dataset highly suitable for the detection of small traffic signs.

In the end, a total of 9457 images were selected, comprising 6598 images in the training set, 1889 images in the validation set, and 970 images in the testing set.

### 4.2. Experimental Environment

The experiment employed the Ubuntu 20.04.3 LTS operating system to establish an efficient experimental environment. The system utilized an Intel(R) Xeon(R) Silver 4314 CPU @ 2.40 GHz with 64 GB RAM, along with an NVIDIA A40 graphics card boasting a memory capacity of 48 GB. Among the available deep learning frameworks, TensorFlow, Keras, and PyTorch stand out as the most prominent choices. For the purpose of efficient training and testing of datasets in this study, the PyTorch deep learning framework was selected. The specific parameters of the experimental environment are outlined in Table 2.

The experiment employed an input image size of 640 × 640 pixels and conducted a total of 300 training epochs. A batch size of 16 was selected, and the SGD optimizer was chosen to expedite the model’s convergence. Detailed experimental parameters for network training can be found in Table 3.

### 4.3. Performance Metrics

In this study, the model was evaluated using three performance metrics: precision, recall, and mean average precision (mAP). Precision represents the probability that the sample that is predicted to be a positive sample is correctly predicted in the prediction result. Recall represents the probability that in the positive sample of the original sample, it will finally be correctly predicted as a positive sample. The formulas for calculating precision and recall are as follows:(8)$$precision=\frac{TP}{TP+FP}$$
(9)$$recall=\frac{TP}{TP+FN}$$
where *TP* (true positive) means that the model predicts positive classes as positive, *FP* (false positive) means that the model predicts negative classes as positive, and *FN* (false negative) denotes that the model predicts positive classes as negative.

*AP* (average precision) refers to the average accuracy in object detection. It combines the model’s performance under different precision and recall conditions and reflects the balance between accuracy and recall. *AP* is calculated as the area under the precision–recall curve (PR curve), as shown in Equation (10):(10)$$AP=\int_{0}^{1}p\left(r\right)dr$$

The *mAP* metric evaluates the overall performance of the model across all categories. It is calculated by taking the average of the *AP* (average precision) values for different categories, as shown in Equation (11):(11)$$mAP=\frac{\Sigma AP}{N_{classes}}$$

### 4.4. Experimental Results and Analysis

#### 4.4.1. Ablation Experiment

To evaluate the impact of our proposed improvements on the performance of traffic sign detection, we conducted a series of ablation experiments. Based on the YOLOv7 baseline model, we added an extra layer for small object detection to YOLOv7, enhancing the model’s ability to perceive small objects, resulting in a 1% increase in mAP. Subsequently, we introduced the ACmix module within the small object detection layer. This module effectively captures small-scale objects and focuses on important regions within the image, leading to a 0.7% improvement in mAP and further enhancing the model’s detection performance. Furthermore, we introduced the omni-dimensional dynamic convolution (ODC) and proposed the ODCBS module, which replaced the original convolutional block Softmax (CBS) module in the neck layer. This replacement yielded a notable 1.2% increase in mAP. Lastly, by introducing the NWD metric in both NMS and the loss function, we addressed the scale sensitivity issue and achieved a significant mAP improvement of 3.1%. Through the validation of our ablation experiments, we demonstrated the positive impact of each improvement on the performance of the traffic sign detection algorithm. In comparison to the baseline model, our SANO-YOLOv7 algorithm achieved significant improvements in terms of precision, recall and mean average precision. The detailed results of the ablation experiments are presented in Table 4.

#### 4.4.2. Performance Comparison

To assess the detection performance of the SANO-YOLOv7 model in comparison to existing object detection models, we conducted evaluations on various recent networks, namely YOLOv3, YOLOv5, YOLOv6, and YOLOv7, as well as two-stage models, including SSD, Faster-RCNN, and those presented by Peng et al. [44] and Cao et al. [45]. The evaluation metrics for each model are summarized in Table 5, providing numerical results for comprehensive comparison.

The proposed SANO-YOLOv7 model demonstrated outstanding overall performance compared to other models. When compared to several other classical target detection algorithms, our SANO-YOLOv7 model exhibited improvements, with mAP@0.5 values of 17.1%, 3.4%, 0.5%, 0.2%, 10.2%, 8.5%, 6.2%, and 5.3%, respectively. These results indicate that the SANO-YOLOv7 model developed in this study surpasses commonly used detection networks in terms of its capability to detect traffic signs. Despite the relatively modest improvement in the recognition accuracy of SANO-YOLOv7 compared to the two-stage model [44,45], the model achieved an approximately five-fold increase in FPS in terms of inference speed, which is of considerable significance for practical engineering applications. In summary, our proposed algorithm strikes a balance between mAP and FPS, achieving relatively higher accuracy while maintaining fast inference speed.

Figure 11 depicts a comparative analysis of the total loss curves for the SANO-YOLOv7 model and other classical target detection models. The graph reveals that the curve of the loss function for the SANO-YOLOv7 model demonstrates higher smoothness, faster stability convergence, and lower values. These characteristics suggest that the SANO-YOLOv7 model exhibited superior training effectiveness and performance.

Figure 12 presents a comparison of the training processes between the SANO-YOLOv7 model and other classical target detection models. This analysis reveals that SANO-YOLOv7 demonstrated robust learning capability during the initial training phase and achieved faster and smoother convergence. Additionally, it yielded higher mAP values. The above comparison shows that the proposed method has good performance.

#### 4.4.3. Dataset Detection Results

To demonstrate the detection capabilities of the SANO-YOLOv7 algorithm for small traffic sign targets, we conducted evaluations by randomly selecting images from the TT100K test set. In this comparative analysis of the dataset’s detection results, we present a comparison between the original YOLOv7 algorithm and our enhanced algorithm when applied to the dataset images. As shown in Figure 13, the first two sets of images clearly demonstrate that our algorithm achieved significant improvements in the accuracy of detecting small traffic sign targets compared to the original YOLOv7 algorithm. Specifically, the accuracy was enhanced by 5%, 19%, 38%, 5%, and 6% for the respective cases. These improvements indicate a notable enhancement in performance. Notably, in the last set of images, while the original YOLOv7 algorithm failed to recognize the target object, our enhanced algorithm successfully detected it. This outcome further substantiates the significant advantages of our improved algorithm in effectively addressing missed detections and enhancing the detection of small targets in the original model.

Through the comparative analysis of these images, it is evident that our enhanced algorithm has achieved remarkable advancements in the accuracy of target object detection when compared to the original YOLOv7 algorithm. These findings offer robust evidence for the reliability and accuracy of our algorithm in real-world applications, establishing a strong basis for further research and practical implementation.

## 5. Discussion

The novelty of this research lies in pioneering the application of the YOLOv7 model for small traffic sign detection, where no previous precedents exist. Our study demonstrates innovation and pioneering efforts in addressing the challenges of small traffic sign detection using the YOLOv7 model. Additionally, we have made improvements to the YOLOv7 model, focusing on overcoming the specific challenges faced in detecting small traffic signs to optimize its performance and conducted tests on the TT100K dataset.

In Section 4.4.1, we discuss the contributions of module additions to the ablation study. As shown in Table 4, incorporating different modules significantly enhanced the network’s detection accuracy. We propose introducing a small object detection layer to improve the original model’s three-layer multi-scale detection structure, effectively enhancing the accuracy of small target detection. The ACmix module utilizes large-scale feature maps from low-level outputs, capturing more features and enhancing the network’s ability to detect smaller objects while reducing noise impact. Furthermore, the ODCBS module employs parallel strategies to learn attentions in all four dimensions of the kernel space, capturing contextual information and prioritizing traffic sign features. Additionally, we introduced the NWD metric as an index to measure the similarity between real and predicted boxes, effectively capturing the spatial information of objects and reducing sensitivity to slight position deviations of small objects, thus significantly improving the accuracy of small object detection.

Based on the comparative experiments in Section 4.4.2, the SANO-YOLOv7 network demonstrated higher precision and speed compared to other mainstream networks. However, when compared to the baseline model YOLOv7, the fusion of multiple modules may increase computational complexity, resulting in a slight decrease in detection speed. Nevertheless, the significant improvement in accuracy while maintaining real-time detection makes this trade-off acceptable. In future research, we will explore the integration of more lightweight modules and residual frameworks into the SANO-YOLOv7 object detection method, aiming to reduce the network’s scale and further enhance its detection accuracy.

## 6. Conclusions

This study proposes the SANO-YOLOv7 algorithm for traffic sign small object detection. The algorithm incorporates several crucial enhancements to the baseline YOLOv7 model. First, to better extract features from small objects, we added a small object detection layer in the neck region, accompanied by an additional small object detection head. Additionally, we utilized the ACmix module, which combines self-attention and convolutional mixing, to capture more features and enhance sensitivity to small-scale targets while reducing the impact of noise. Furthermore, we introduced the omni-dimensional dynamic convolution and proposed the ODCBS module, replacing the original CBS module in the neck layer, aiming to capture contextual information and prioritize traffic sign features. Lastly, we introduced the normalized Gaussian Wasserstein distance (NWD) metric to mitigate the sensitivity of IoU loss to the positional deviations of small objects. The experimental results obtained from the TT100K dataset clearly show that our proposed method effectively addresses the need for detecting small traffic signs. Moreover, our method ensures real-time detection with a notable enhancement in detection accuracy. 

However, this study has certain limitations, mainly related to the research scope and dataset restrictions. Initially, our focus was solely on classifying and recognizing traffic signs in our country, without considering traffic signs from other countries or regions. As traffic signs can vary across different locations, future research should expand its scope to include traffic signs from more diverse regions, thereby enhancing the model’s ability to generalize. Additionally, despite our efforts to collect a substantial number of traffic sign data samples, the dataset’s diversity and quantity remain limited. This limitation could potentially affect the model’s training and overall performance. To enhance the model’s accuracy and robustness, future research should consider gathering more representative traffic sign samples and employing data augmentation techniques to increase the dataset’s diversity. Finally, it is crucial to pursue model lightweighting to facilitate deployment on mobile devices, thereby improving the algorithm’s practicality.

## Figures and Tables

**Figure 1 sensors-23-07145-f001:**
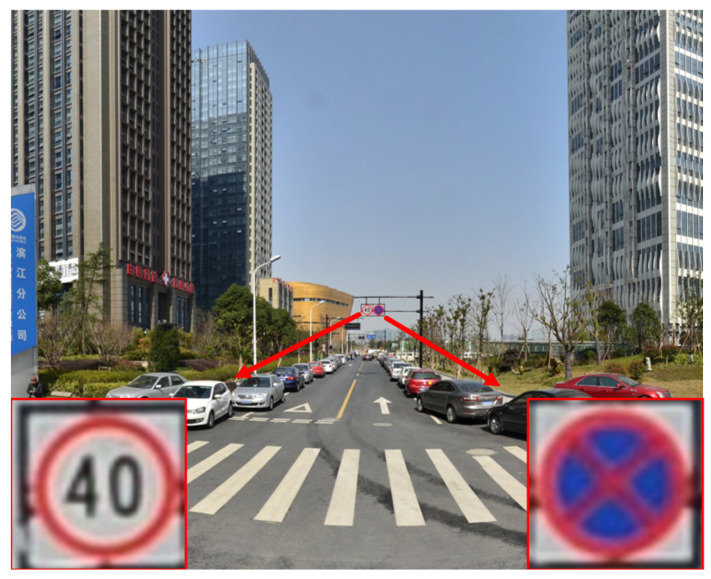
Small traffic signs in the image.

**Figure 2 sensors-23-07145-f002:**
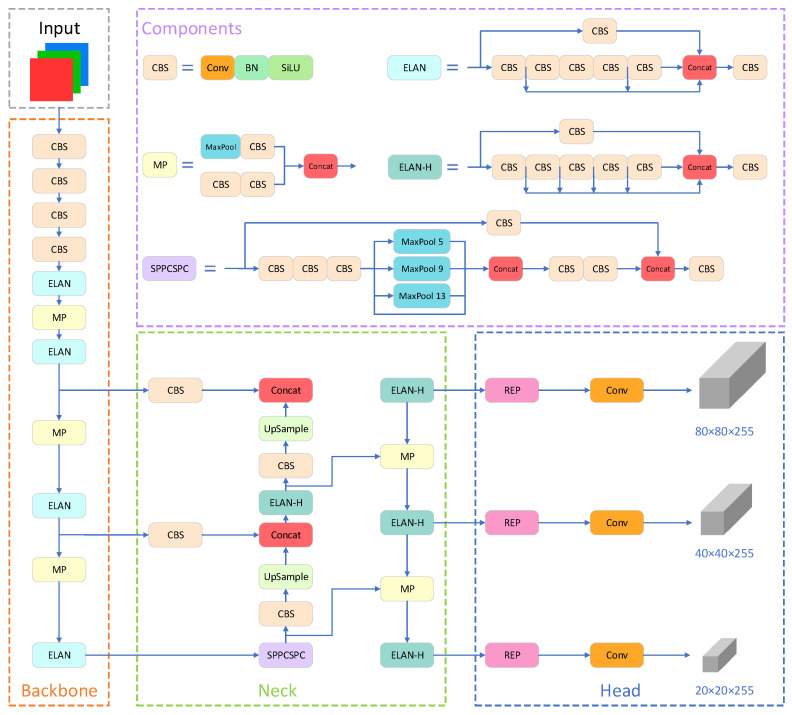
Architecture of YOLOv7 network.

**Figure 3 sensors-23-07145-f003:**
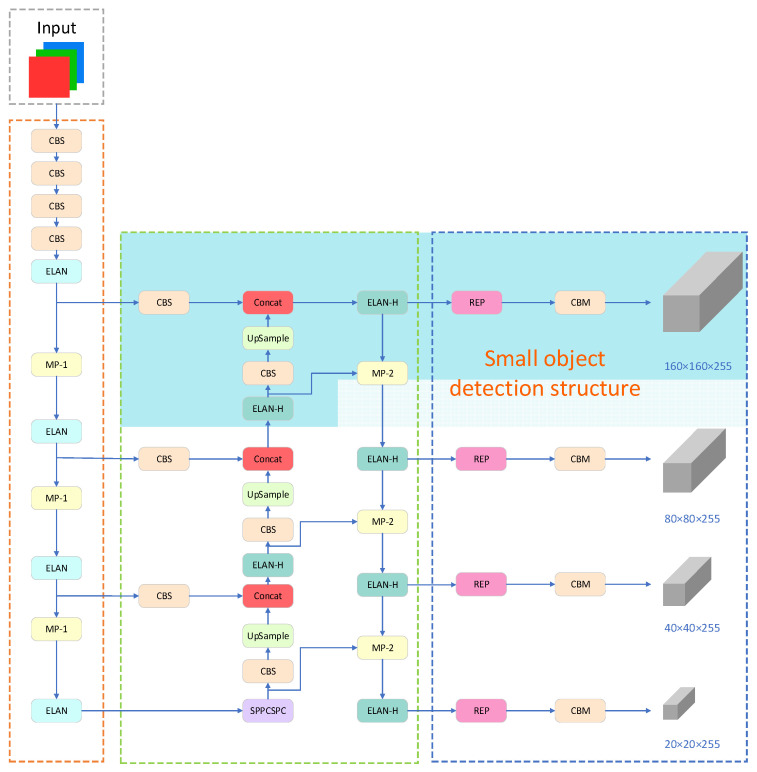
Small object detection structure.

**Figure 4 sensors-23-07145-f004:**
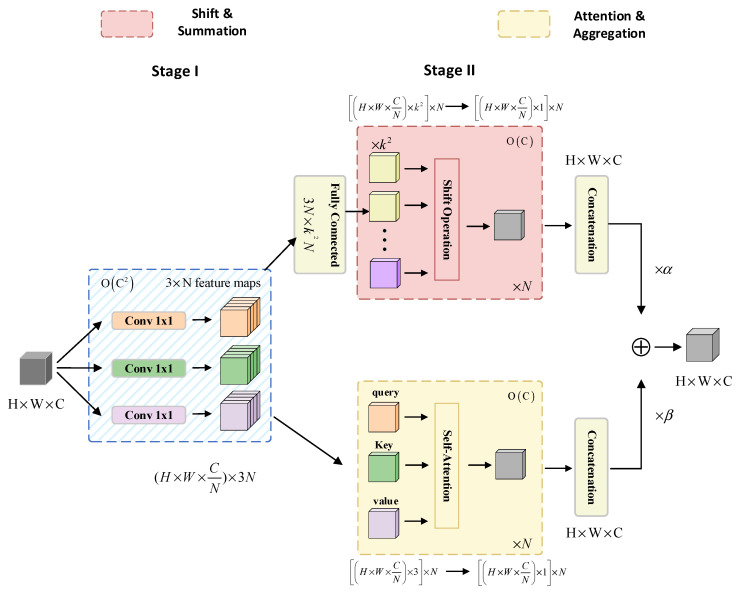
The structure of ACmix.

**Figure 5 sensors-23-07145-f005:**
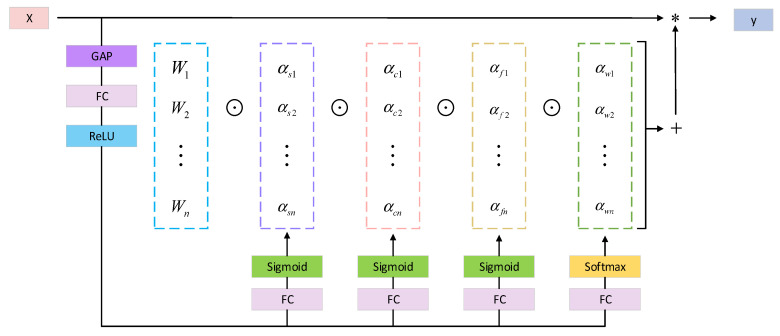
Calculation process of the ODConv.

**Figure 6 sensors-23-07145-f006:**
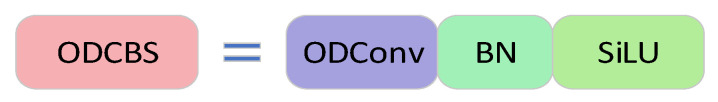
ODCBS structure diagram.

**Figure 7 sensors-23-07145-f007:**
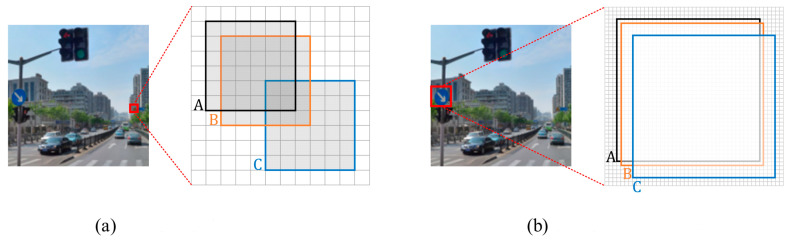
The sensitivity analysis of IoU on tiny and normal scale objects. (**a**) Tiny scale object; (**b**) Normal scale object.

**Figure 8 sensors-23-07145-f008:**
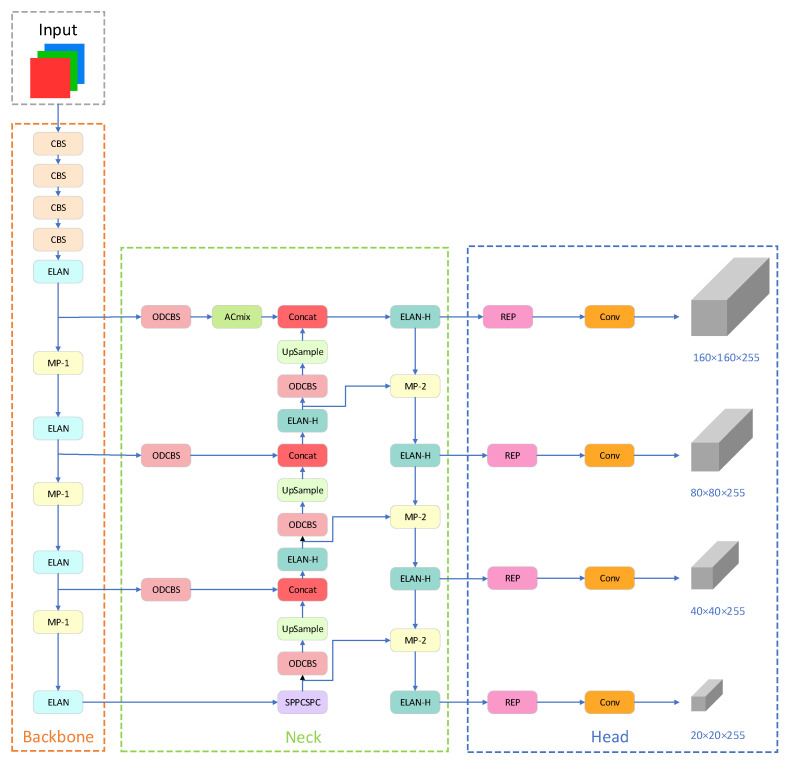
The network architecture of SANO-YOLOv7.

**Figure 9 sensors-23-07145-f009:**
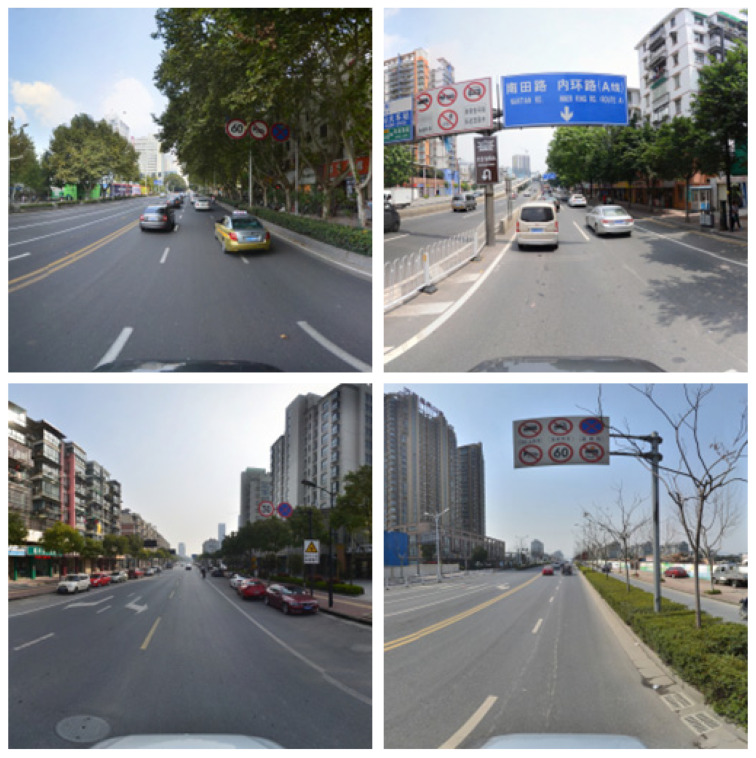
Sample images from the TT100K dataset.

**Figure 10 sensors-23-07145-f010:**
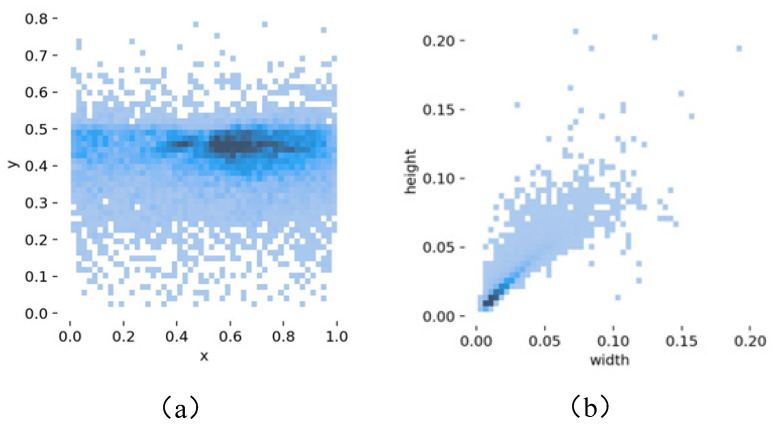
Dataset analysis results. (**a**) Location distribution of dataset target center point. (**b**) Distribution of dataset target size.

**Figure 11 sensors-23-07145-f011:**
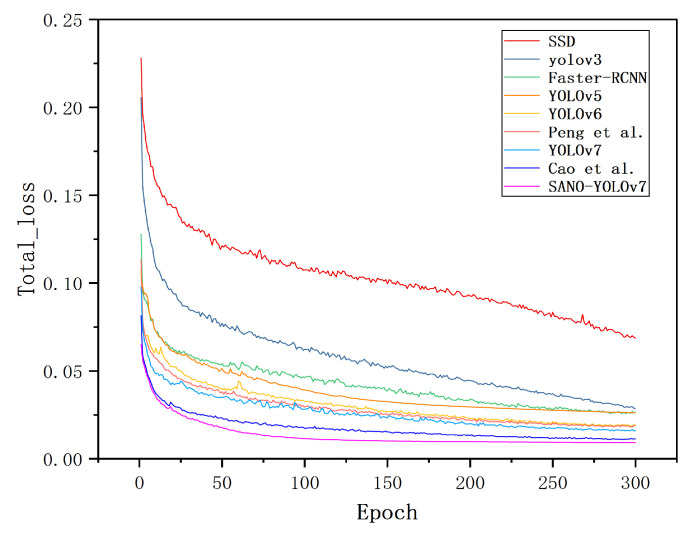
Comparison chart of loss function of SANO-YOLOv7 and other classical target detection models (including Peng et al. [44] and Cao et al. [45]).

**Figure 12 sensors-23-07145-f012:**
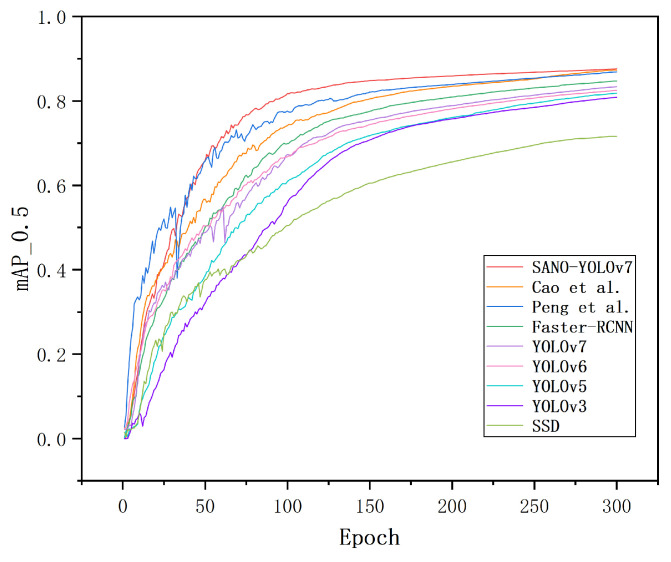
Comparison chart of mAP of SANO-YOLOv7 and other classical target detection models (including Peng et al. [44] and Cao et al. [45]).

**Figure 13 sensors-23-07145-f013:**
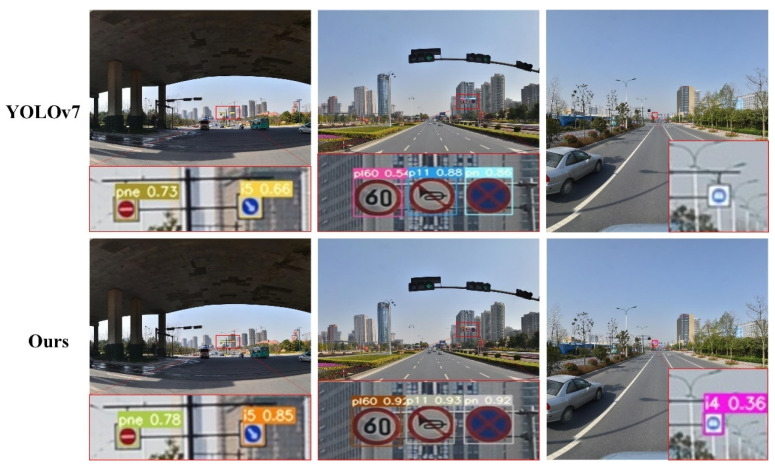
Comparison chart of YOLOv7 and our method.

**Table 1 sensors-23-07145-t001:** The 42 types of traffic signs used in the experiment.

Category	Sign Name
Warning signs	w13, w55, w57, w59
Prohibition signs	p10, p11, p12, p19, p23, p26, p27, p3, p5, p6, pg, ph4, ph4.5, pl100, pl120, pl20, pl30, pl40, pl5, pl50, pl60, pl70, pl80, pm20, pm30, pm55, pn, pne, po, pr40
Directional signs	i2, i4, i5, il100, il60, il80, io, ip

**Table 2 sensors-23-07145-t002:** Experimental environmental parameters.

Component	Name/Value
Operating system	Ubuntu 20.04.3 LTS
CPU	Intel(R) Xeon(R) Silver 4314 CPU@ 2.40 GHz
GPU	NVIDIA A40
Video memory	48 GB
Training acceleration	CUDA 11.7
Programming language	Python 3.9
Deep learning framework for training	PyTorch 1.13.1

**Table 3 sensors-23-07145-t003:** Experimental parameters of network training.

Component	Name/Value
Input image size	640 × 640 pixels
Epoch	300
Training batch size	16
Initial learning rate	0.01
Final learning rate	0.1
Momentum	0.937
Weight_decay	0.0005
Optimizer	SGD

**Table 4 sensors-23-07145-t004:** Ablation experiment based on baseline.

Methods	Multiscale Small Object Detection Structure	ACmix	ODCBS	NWD	Precision	Recall	mAP@0.5(%)	mAP@0.5:0.95(%)
YOLOv7					81.4	75.3	83.4	66.1
	√				85	74.9	84.4	67.1
			√		85	75	84.6	67.2
				√	84.7	78.1	86.5	68.7
	√	√			84.2	77.4	85.1	68.1
	√		√		84.1	77.7	85.3	68.2
	√			√	85.4	80.3	87.6	69
	√	√	√		84.8	77.9	86.2	67.9
	√	√		√	87	80	88.2	69.9
	√		√	√	87.1	79.9	88.1	69.9
	√	√	√	√	87.1	80.1	88.7	70.5

**Table 5 sensors-23-07145-t005:** Comparison of different models in experiments.

Methods	P (%)	R (%)	mAP@0.5 (%)	FPS	Param (M)
SSD	70.6	77.2	71.6	55	101
Faster-RCNN	80.8	81	85.3	20	150
Peng et al. [44]	88	80.5	88.2	18	155.6
Cao et al. [45]	87.5	81	88.5	19	153.4
YOLOv3	69.2	78.1	78.5	72	61
YOLOv5	72.8	81.4	80.2	87	20
YOLOv6	74.5	83.7	82.5	90	42
YOLOv7	83.2	74.4	83.4	107	35.4
SANO-YOLOv7	87.1	80.1	88.7	90	35.7

## Data Availability

Not applicable.

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
