# Peer review of "A Small Object Detection Algorithm for Traffic Signs Based on Improved YOLOv7"

_sensors, 2023, doi:10.3390/s23167145_

Round 1

Reviewer 1 Report

The manuscript provides a comprehensive overview of the proposed SANO-YOLOv7 algorithm for traffic sign small object detection. The introduction effectively sets the context for the research and includes relevant references. However, I would suggest the authors revise the introduction to provide more clarity and engage the reader from the beginning.  The results are clearly presented, and the authors have effectively compared the performance of the SANO-YOLOv7 model with other existing object detection models.

[line 42]
It would be better to provide an example or more detailed explanation for a small target traffic sign.

[line 40-43 and 52-54]
How to connect challenges and the objective? It should be more elaborated. The authors claimed that the traffic sign detection technology encounters challenges in detecting small target objects, and the use of TT100k and GTSRB amplifies the challenge of detecting small targets. Then logically, readers can think that the existing open datasets have a problem detecting the small target object, and the existing open datasets should be changed. However, the authors are presenting
the new algorithm only and conducting an experiment using the same dataset (TT100K) even though the existing dataset amplifies the challenge. The authors should more clearly clarify the problem statement, and the problem statement should be logically connected to the objective.

[line 48]
It would be better if prior works to address the challenges were provided.

  1. Related Works
    The authors should delve into the shortcomings of some algorithms in 2.2 Small object detection [line 184], and the authors should show the reason why they used the YOLOv7 by emphasizing the pros of your algorithm compared to other algorithms in 2.3 YOLOv7 Network Structure [line 222].

  2. MaterialsandMethods
    [line 285-287]
    Is this sentence a subjective opinion? If it is, please put the reason why these only two primary challenges should be addressed. If it is from any other sources, please put the citation.

  3. ExperimentsandAnalysisofResults
    [line 593-595]
    The authors need to add a discussion part. For example, how will you explain the FPS comparison between YOLOv7 and SANO-YOLOv7? The line 605-607 can be included in the discussion part.

  4. Conclusion
    What is the limitation of this paper? The limitation of this paper needs to be mentioned in the conclusion. For example, this study utilized TT100K dataset for the ablation experiment. If the GTSRB is used for the experiment, the result may change. Such limitations can be included in the Conclusion. 

Minor revision is recommended. 

Reviewer 2 Report

The paper is a study on the small object detection algorithm for traffic signs based on improved YOLOv7 and is considered a valuable and interesting study in related fields. The reviewer's opinions are as follows.

1. Abstract should be concisely and clearly described, including the background, purpose, method, result, and conclusion of the study.

2. In the description, ambiguous expressions should be avoided and quantitative numerical values or objective grounds should be presented. 

3. It is necessary to describe existing efforts(papers) regarding the problems (not the simple description of the existing studies). The methods that solved the problems perceived in previous similar studies should be described in detail(academic excellence on this paper).

4. In the section describing the simulations and experiments, the composition of the simulations and datasets should be clearly explained. It should be described in such a way that readers who related the fields can understand processes in detail. In other words, it should be possible to solve the questions by the composition of the provided experimental environments including the models suggested and datasets provided. In the simulation part, verification of the proposed methodology should be sufficiently presented.

5. In the 'conclusion' part, it is necessary to describe the limitations of the study and additional studies required in the future. It is recommended to describe the interpretation of the research results in an easy-to-understand manner. 

6. Authors should ensure the quality of the paper in its overall descriptive expression and follow the format of the journal.

Thank you very much.

Authors should ensure the quality of the paper in its overall descriptive expression and follow the format of the journal.

Thank you very much.

Author Response

请参阅附件。

Reviewer 3 Report

1.Please provide the innovation of the SANO-YOLOv7 algorithm.

2.Please compare the performance with Faster RCNN algorithm, such as "An Improved Faster R-CNN for Small Object Detection "in IEEE Access instead of [14] in 2015.

3.Suggest adding algorithm comparisons for Figure 10 and Figure 11, rather than just two algorithms.

Moderate editing of English language required。

Author Response

请参阅附件。

Reviewer 4 Report

The paper is introduced the new SANO-Yolov7 algorithm for small target detection, which builds upon the Yolov7 framework.

The title, abstract, and introduction were found appropriate. It is positive that the main contribution and the general organization of the article are presented, especially in the introduction.

The authors have provided qualitative comparisons with previous studies, which is useful for assessing the scientific and practical novelty of the study.

The results of all tables are explained in detail. Quantitative indicators of comparison with existing approaches are also provided. This makes it clear the advantages of the proposed method over the existing ones.

The paper is correctly structured.

The paper contains significant theoretical results and new and significant experimental results.

I believe that this work can be accepted in the form presented.

Author Response

请参阅附件。

Round 2

Reviewer 3 Report

The comparison of algorithm performance can also be improved, not only through the use of open source code, but also as a reference for performance indicators.
